# SUBJECTIVE NEURAL NETWORKS: BAYESIAN DROPOUT WITH TRUST-AWARE OPINIONS

## ABSTRACT

Deep neural networks achieve remarkable predictive accuracy but often fail to convey meaningful uncertainty, which limits their reliability in safety-critical applications. Existing approaches such as Evidence Deep Learning (EDL) and Bayesian Dropout either treat uncertainty as deterministic evidence or approximate it with sampling, but they lack an explicit interpretation of subjective trust. In this work, we introduce the Subjective Neural Network (SNN), a framework that combines Bayesian variational inference with subjective logic. Neuron activations are controlled by Beta–Bernoulli Dropout, where the Beta distribution encodes a subjective trust opinion and the Bernoulli mask determines whether a neuron participates in inference. During prediction, we apply a nested sampling procedure: sampling trust probabilities from Beta distributions, generating Dropout masks, and aggregating outputs into Dirichlet distributions. This process produces predictions that can be directly mapped into subjective opinions of beliefs and uncertainty over class labels. Empirical results on image classification benchmarks show that SNN achieves competitive accuracy while providing calibrated and interpretable uncertainty estimates. Our work establishes a principled connection between Bayesian deep learning and subjective logic, offering a pathway toward trust-aware neural networks.

## 1 INTRODUCTION

Neural Networks (NNs) have achieved state-of-the-art performance across a wide range of tasks, from image recognition to natural language processing. Despite these successes, their limited ability to communicate meaningful uncertainty remains a critical bottleneck for deploying them in safety-sensitive domains such as healthcare, autonomous driving, and security systems. While predictive accuracy is indispensable, the lack of reliable measures of uncertainty can lead to overconfidence in incorrect predictions, undermining trust and hindering adoption in high-stakes applications.

A rich body of research has attempted to address this challenge. Bayesian neural networks (BNNs) offer a principled foundation by modeling distributions over weights, but they often suffer from scalability and computational overhead. Variants of dropout have been reinterpreted as approximate Bayesian inference, providing practical uncertainty estimates, yet these methods still conflate ignorance with stochastic variability. Evidence-based approaches such as Evidential Deep Learning (EDL) instead introduce Dirichlet evidence distributions, enabling models to express "I do not know" when faced with ambiguous or out-of-distribution inputs. However, both Bayesian Dropout and EDL fall short of explicitly encoding *subjective trust*, an essential component for reasoning under uncertainty.

Subjective Logic (SL) offers a natural formalism for addressing this missing dimension. Beyond modeling predictive uncertainty, SL allows uncertainty itself to be structured into interpretable opinions that distinguish between beliefs and residual ignorance. In the binary case, this reduces to belief, disbelief, and uncertainty, which provides a natural way to represent and reason about trust. Prior work has demonstrated its value in tasks such as dataset and neural network trustworthiness assessment. Yet, its integration with deep learning remains limited, and a unified framework that combines Bayesian inference with subjective trust reasoning has not been fully explored.

In this work, we propose the *Subjective Neural Network (SNN)*, a framework that bridges Bayesian Dropout with subjective logic. Our approach introduces Beta–Bernoulli dropout as a trust-aware

mechanism at the neuron level, where Beta distributions encode subjective trust and Bernoulli masks govern neuron participation. Predictions are aggregated into Dirichlet distributions, which directly map into subjective opinions over class outcomes. This design allows SNNs to retain the computational efficiency of variational dropout while providing interpretable and calibrated uncertainty estimates grounded in subjective trust.

Our contributions are as follows:

1. A principled integration of Bayesian variational inference and subjective logic, enabling trust-aware uncertainty modeling in neural networks.

2. A novel generative mechanism via Beta–Bernoulli dropout, where neuron activations are governed by trust distributions.

3. An inference procedure that produces Dirichlet-based opinions, yielding predictions with explicit separation between belief and ignorance.

4. Extensive empirical validation on MNIST and CIFAR benchmarks, demonstrating that SNNs achieve competitive accuracy while improving calibration, robustness to covariate shift, and resilience under adversarial attacks.

Together, these results establish SNNs as a step toward interpretable, trust-aware deep learning, aligning predictive power with the demands of reliable decision-making.

## 2 UNCERTAINTY QUANTIFICATION AND EVIDENCE-BASED DEEP LEARNING

**Uncertainty in Deep Learning.** Quantifying predictive uncertainty is a central challenge in deep learning. Bayesian neural networks (BNNs) (Neal, 1996; Blundell et al., 2015) model weight distributions explicitly, but are computationally expensive in practice. Dropout has been reinterpreted as approximate Bayesian inference (Gal & Ghahramani, 2016b), providing a scalable alternative for uncertainty estimation. Several extensions improve its flexibility: variational dropout (Molchanov et al., 2017) learns individual dropout rates, and Concrete Dropout (Gal et al., 2017; Maddison et al., 2017) introduces a differentiable relaxation for optimizing dropout probabilities. While these methods capture both model and data uncertainty, they ultimately express uncertainty only as variance in probability estimates and do not explicitly separate belief from ignorance. Beta–Bernoulli Dropout, introduced by Lee et al. (2019), places a Beta prior over dropout probabilities, enabling each neuron to evolve as generic, specialized for certain inputs, or dropped entirely. This approach is primarily used for network sparsification rather than for uncertainty quantification.

**Subjective Logic.** Subjective Logic (SL) (Jø sang, 2016) extends Dempster–Shafer theory (Dempster, 1967; Shafer, 1976) to represent beliefs as *opinions* rather than fixed probabilities. A multinomial opinion over $K$ outcomes is

$$\omega = (\mathbf{b}, u, \mathbf{a}),$$

with belief masses $\mathbf{b} = (b_1, \ldots, b_K)$, uncertainty $u$, and base rate prior $\mathbf{a}$. Conventional probabilistic models capture only *aleatoric* uncertainty arising from inherent randomness, whereas SL also makes *epistemic* uncertainty explicit through the quantity $u$, accounting for limited knowledge or missing evidence. From an opinion, one can recover a projected probability distribution via

$$p_k = b_k + u \times a_k, \ \forall k \in \{1, 2, \ldots, K\}$$

which incorporates both committed belief and residual uncertainty. This representation of opinion maps to a Dirichlet distribution with concentration parameters $\boldsymbol{\eta} = (\eta_1, \ldots, \eta_K)$:

$$b_k = \frac{\eta_k}{S + K}, \quad u = \frac{K}{S + K}, \quad a_k = \frac{1}{K}, \qquad S = \sum_k \eta_k,$$

where $\eta_k$ represents the amount of *evidence* supporting outcome $k$. A special case of the multinomial opinion is the binomial opinion ($K = 2$), which maps to a Beta distribution and is often used to model trust in the occurrence of an event.

Beyond its theoretical appeal, SL has been applied in AI to assess the trustworthiness of datasets (Ouattara et al., 2025b) and to evaluate neural network trustworthiness through evidence-based trust measures (Cheng et al., 2020; Ouattara et al., 2025a).

**Evidential Deep Learning.** Building on the connection between opinions and Dirichlet distributions, Sensoy et al. (2018) proposed *Evidential Deep Learning* (EDL), where neural networks output non-negative evidence values that define Dirichlet parameters and corresponding opinions. The model is trained with a loss that encourages high evidence for correct predictions and low evidence when the input cannot be confidently explained, effectively allowing the network to express "I do not know." This yields calibrated uncertainty estimates and improves performance in tasks such as out-of-distribution detection and adversarial robustness, without requiring Monte Carlo sampling.

**Motivation.** Bayesian methods provide principled approximate inference, while subjective logic and EDL offer interpretable decompositions of uncertainty into belief and ignorance. Our goal is to unify these perspectives into a single framework, the *Subjective Neural Network (SNN)*, which enables reasoning about trust directly within the network. This offers improved interpretability and better-calibrated predictions, while keeping predictive performance competitive.

## 3 SUBJECTIVE NEURAL NETWORKS

We propose the *Subjective Neural Network* (SNN), a Bayesian neural architecture where uncertainty is represented through subjective trust opinions. The key idea is to model dropout probabilities as latent random variables drawn from Beta distributions, interpreted as subjective trust in the activation of neurons. Predictions are then aggregated into Dirichlet distributions, which correspond to subjective opinions over class outcomes.

### 3.1 GENERATIVE MODEL VIA BETA–BERNOULLI DROPOUT

For each neuron in the network, we introduce a latent trust parameter

$$p_j \sim \text{Beta}(\alpha_j, \beta_j),$$

which reflects the subjective belief about whether neuron $j$ should contribute to inference (which can also be interpreted as trust in that neuron). Conditioned on $p_j$, we sample a Bernoulli mask

$$z_j \sim \text{Bernoulli}(p_j),$$

and apply it to the neuron's activation. This mechanism extends classical dropout: instead of a fixed rate, each neuron has a distribution over dropout probabilities, capturing uncertainty.

The forward pass for input $x$ under a sampled mask $z$ is therefore

$$f(x; W, z) = \text{NN}(x; W \odot z),$$

where $W$ are weights and $z$ are Bernoulli samples conditioned on Beta-distributed $p$.

### 3.2 VARIATIONAL INFERENCE WITH BETA DISTRIBUTIONS

Direct inference of the posterior over trust probabilities is intractable, so we adopt a variational approach. We approximate each $p_j$ with a variational Beta distribution

$$q(p_j) = \text{Beta}(\alpha_j, \beta_j),$$

while the prior is set to $p(p_j) = \text{Beta}(a_0, b_0)$. We set $a_0 = b_0 = 1$ to use a uniform prior.

**Kumaraswamy reparameterization.** To enable gradient-based learning, we employ the Kumaraswamy distribution (Mitnik & Baek, 2013), which closely approximates the Beta while being reparameterizable:

$$p_j = \left(1 - u^{\frac{1}{\beta_j}}\right)^{\frac{1}{\alpha_j}}, \qquad u \sim \text{Uniform}(0, 1).$$

This allows differentiable sampling of $p_j$ with respect to $(\alpha_j, \beta_j)$.

---

**Algorithm 1** Training a Subjective Neural Network

---

1: Initialize variational parameters $\{\alpha_j, \beta_j\}$ for all neurons.
2: **for** each minibatch $(x, y) \sim D$ **do**
3:    **for** each neuron $j$ **do**
4:       Sample $u \sim \text{Uniform}(0, 1)$
5:       Reparameterize $p_j \leftarrow (1 - u^{1/\beta_j})^{1/\alpha_j}$
6:       Sample mask $z_j \sim \text{Bernoulli}(p_j)$ (or use Concrete relaxation)
7:    **end for**
8:    Forward pass: $\hat{y} \leftarrow f(x; W, z)$
9:    Compute ELBO:
$$\mathcal{L} = \log P(y \mid \hat{y}) - \sum_j \text{KL}[q(p_j) \,\|\, p(p_j)]$$

10:    Update $\alpha_j, \beta_j$ via backpropagation
11: **end for**

---

**ELBO Objective.** The variational training objective is the evidence lower bound (ELBO):
$$\mathcal{L} = \mathbb{E}_{q(p)}\Big[ \log P(D \mid W(p)) \Big] - \text{KL}[q(p) \,\|\, p(p)],$$

where $W(p)$ denotes weights masked by Bernoulli draws conditioned on $p$. The KL divergence between two Beta distributions has a closed form:

$$
\begin{aligned}
\text{KL}(\text{Beta}(\alpha_j, \beta_j) \,\|\, \text{Beta}(a_0, b_0)) = {} & \log \frac{B(a_0, b_0)}{B(\alpha_j, \beta_j)} + (\alpha_j - a_0)\psi(\alpha_j) \\
& + (\beta_j - b_0)\psi(\beta_j) - (\alpha_j + \beta_j - a_0 - b_0)\psi(\alpha_j + \beta_j),
\end{aligned}
$$

where $B(\cdot, \cdot)$ is the Beta function and $\psi(\cdot)$ the digamma function. The overall training procedure is summarized in Algorithm 1.

### 3.3 Inference via Nested Sampling

During inference, we approximate the posterior predictive distribution by *nested sampling*. We first draw multiple samples of trust probabilities $p$, and for each, draw multiple Bernoulli masks:

1. For $i = 1, \ldots, N_p$: sample $p^{(i)} \sim q(p)$.

2. For $j = 1, \ldots, N_m$: sample $z^{(i,j)} \sim \text{Bernoulli}(p^{(i)})$.

3. Compute predictive probabilities
$$\pi^{(i,j)} = \text{Softmax}(f(x; W, z^{(i,j)})).$$

4. Average across masks to obtain one probability vector per Beta sample:
$$\bar{\pi}^{(i)} = \frac{1}{N_m} \sum_{j=1}^{N_m} \pi^{(i,j)}.$$

Collecting the set $\{\bar{\pi}^{(i)}\}_{i=1}^{N_p}$ yields a distribution of probability for each class, which we approximate with a Dirichlet by minimizing the KL divergence to the empirical distribution. This corresponds to fitting Dirichlet parameters $\hat{\boldsymbol{\eta}}$ as following:
$$\hat{\boldsymbol{\eta}} = \arg\min_{\boldsymbol{\eta}} \ \text{KL}\big(\hat{p}(\pi) \,\|\, \text{Dir}(\pi \mid \boldsymbol{\eta})\big),$$

These parameters are then mapped into a multinomial opinion:
$$b_k = \frac{\hat{\eta}_k}{\hat{S} + K}, \quad u = \frac{K}{\hat{S} + K}, \quad a_k = \frac{1}{K}, \qquad \hat{S} = \sum_k \hat{\eta}_k.$$

Thus, the output of the SNN for input $x$ is not just a probability vector, but a structured subjective opinion that explicitly captures both belief and residual uncertainty. The full inference procedure is summarized in Algorithm 2.

---

**Algorithm 2** Inference with a Subjective Neural Network

---

1: Given trained parameters $\{\alpha_j, \beta_j\}$ and input $x$
2: **for** $i = 1 \ldots N_p$ **do**
3:     Sample trust parameters $p^{(i)} \sim \text{Beta}(\alpha_j, \beta_j)$
4:     **for** $j = 1 \ldots N_m$ **do**
5:         Sample masks $z^{(i,j)} \sim \text{Bernoulli}(p^{(i)})$
6:         Compute $\pi^{(i,j)} = \text{Softmax}(f(x; W, z^{(i,j)}))$
7:     **end for**
8:     Average over masks to obtain one probability vector per Beta sample:

$$\bar{\pi}^{(i)} = \frac{1}{N_m} \sum_{j=1}^{N_m} \pi^{(i,j)}$$

9: **end for**
10: Collect the set $\{\bar{\pi}^{(i)}\}_{i=1}^{N_p}$ for each class, and approximate to a Dirichlet Distribution
11: Fit Dirichlet parameters $\hat{\eta}$ and map to a multinomial opinion.

---

## 4 EXPERIMENTS

We conduct a comprehensive evaluation of Subjective Neural Networks (SNNs) against Monte Carlo Dropout (Gal & Ghahramani, 2016a) (referred to as Dropout) and Evidential Deep Learning (EDL) (Sensoy et al., 2018). For Dropout, we use the best-performing variant identified in Gal & Ghahramani (2016a), with dropout applied after every parameter layer.

Our experiments are designed to assess both predictive performance and the quality of uncertainty estimation. Following the protocol of Sensoy et al. (2018), we evaluate models across four dimensions: (i) standard classification accuracy, (ii) calibration and the tradeoff between accuracy and uncertainty, (iii) out-of-distribution (OOD) detection, and (iv) robustness to adversarial perturbations. This suite of tests provides a balanced view of accuracy, calibration, and interpretability, allowing us to highlight the advantages of trust-aware uncertainty in SNNs.

We evaluate on MNIST (10 classes, OOD: notMNIST) and CIFAR-5 (first five CIFAR-10 classes, OOD: last five classes). For MNIST we use a LeNet-style CNN with two convolutional layers (20 and 50 filters, kernel size $5 \times 5$) each followed by max-pooling, a dense layer with 500 units, and a final softmax output layer. For CIFAR-5 we use a CNN with two convolutional layers (32 and 64 filters, kernel size $5 \times 5$) each followed by max-pooling, a dense layer with 1000 units, and a final softmax output layer.

All models are implemented in Python using TensorFlow. Optimizer is Adam (lr=$10^{-3}$), batch size 256, for 50 epochs on MNIST and 100 on CIFAR-5. SNNs are trained with variational inference using the Kumaraswamy reparameterization as illustrated in the previous section. At test time, SNNs use $N_p = 10$ Beta samples and $N_m = 10$ Bernoulli masks, while MC Dropout uses 100 forward passes (corresponding to $N_p \times N_m$) forward passes. Adversarial robustness is evaluated with FGSM perturbations $\epsilon \in [0, 0.4]$. We compare against MC Dropout (Gal & Ghahramani, 2016b) and EDL (Sensoy et al., 2018).

Table 1 reports test accuracies on MNIST and CIFAR-5. SNN achieves accuracy comparable to Dropout and better than EDL on MNIST, while on CIFAR-5 it improves over EDL and remains competitive with Dropout. In both cases, SNN provides better-calibrated uncertainty estimates, as shown in later sections.

Table 1: Test accuracies (%) for MNIST and CIFAR-5.

| Method | MNIST | CIFAR-5 |
|---|---|---|
| Dropout | 99.35 | 81.58 |
| EDL | 98.42 | 71.24 |
| **SNN (ours)** | **99.21** | **76.21** |

## 4.1 EVALUATION ON ROTATED IMAGES

We further test robustness to covariate shift by evaluating models on rotated digits from MNIST. Instead of plotting the belief, we plot the projected probability in order to compare with the Dropout model. Figure 1 illustrates predictions for digits 1 and 9 under rotations between 0° and 180°. All methods maintain correct predictions at small angles, but their uncertainty behaviors differ. EDL often assigns higher uncertainty when misclassifying (e.g., predicting 7 instead of 1), while Dropout and SNN tend to remain overconfident. For digit 9, predictions gradually shift to 6 as expected with rotation; however, EDL occasionally predicts 5 and SNN sometimes predicts 2, both with elevated uncertainty, whereas Dropout produces more scattered misclassifications with lower uncertainty. When examining belief rather than projected probabilities, both SNN and EDL tend to assign lower belief than uncertainty when making incorrect predictions, reflecting a cautious response to errors. Interestingly, even at the level of projected probabilities, SNN can remain more robust, sometimes avoiding misclassifications entirely across all rotation angles (e.g., the CIFAR-5 object of class 4 in Appendix Figure 5).

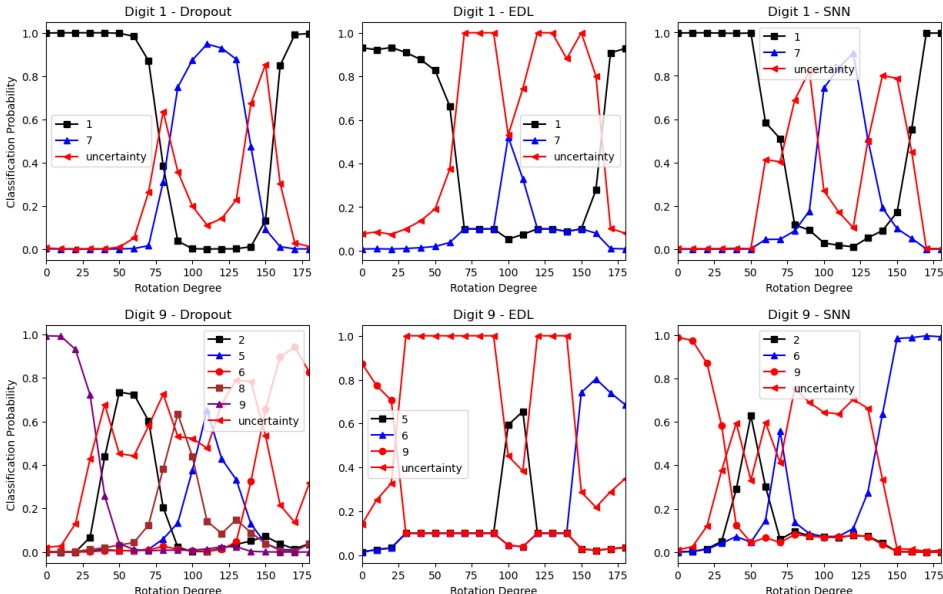

Figure 1: Classification of rotated digits 1 and 9 at different angles (0°–180°) for Dropout, EDL, and SNN.

## 4.2 PREDICTIVE UNCERTAINTY PERFORMANCE

We first evaluate how predictive accuracy varies with uncertainty thresholds. In this setting, predictions whose uncertainty exceeds a chosen threshold are rejected, and accuracy is computed only on the retained predictions. This provides a measure of how well uncertainty estimates align with correctness. A well-calibrated model should achieve higher accuracy as the threshold tightens, approaching 100% when only its most confident predictions are considered. Figure 2 shows accuracy as a function of uncertainty threshold on MNIST. As expected, accuracy increases as high-uncertainty predictions are rejected, converging to nearly 100% when only the most confident predictions are retained. Both SNN and EDL follow this trend, but EDL exhibits an abrupt accuracy drop from ≈100% to 87% at high thresholds, suggesting discontinuous calibration. In contrast, SNN decreases more smoothly, indicating a more stable uncertainty–accuracy relationship.

Figure 3 reports the empirical CDF of predictive entropy for notMNIST (OOD) versus MNIST (in-distribution). The cumulative distribution function (CDF) shows the proportion of samples whose entropy falls below a given threshold, providing a way to compare how uncertainty is distributed across methods. Entropy values lie in the range $[0, \log(10)]$, with curves closer to the bottom right indicating higher entropy for OOD samples. EDL achieves the strongest overall separation, but SNN

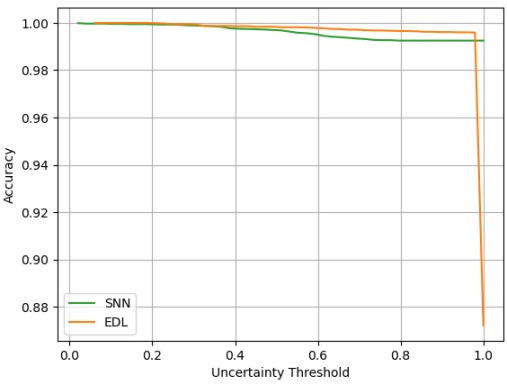
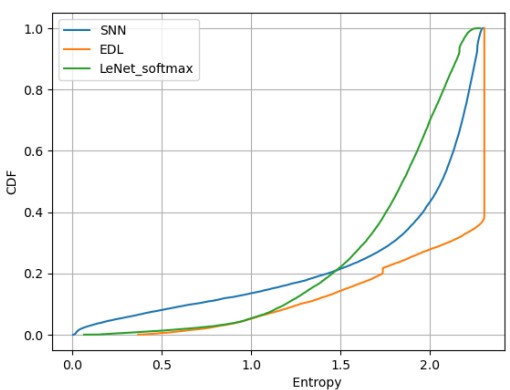

Figure 2: Accuracy as a function of uncertainty threshold on MNIST for Dropout, EDL and SNN.

Figure 3: Empirical CDF for the entropy of the predictive distributions on the notMNIST.

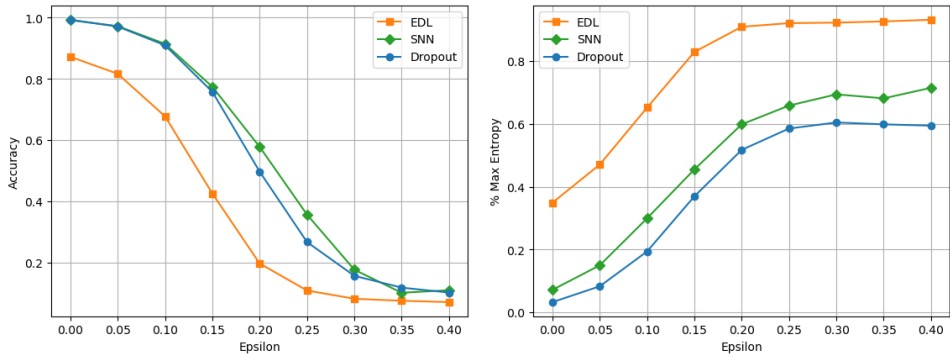

Figure 4: Accuracy and entropy as a function of the adversarial perturbation $\epsilon$ on the MNIST dataset.

also surpasses Dropout after around entropy $\approx 1.5$, showing that SNN can assign higher uncertainty to more than 80% of the OOD inputs. This behavior reflects the model's ability to remain conservative in borderline cases, providing complementary strengths to EDL's sharper separation. We also note that while EDL generally provides stronger OOD uncertainty estimates, it can occasionally assign near-maximal uncertainty even to in-distribution samples (as seen in Figure 2), highlighting a limitation of its calibration.

### 4.3 ADVERSARIAL ROBUSTNESS

We further evaluate our approach under adversarial perturbations generated with the Fast Gradient Sign Method (FGSM) implemented in TensorFlow following the design of CleverHans (Papernot et al., 2018). For each model trained in the previous experiments, adversarial examples are crafted by adding perturbations of varying strength $\epsilon \in [0, 0.4]$ to the input images. As $\epsilon$ increases, examples move further from the data manifold, making prediction progressively more difficult. We use these adversarial examples to probe both accuracy and predictive uncertainty.

Figure 4 shows the evolution of accuracy and normalized entropy on MNIST. All methods experience reduced accuracy with larger $\epsilon$, as expected. SNN maintains accuracy comparable to Dropout and EDL while exhibiting a steady rise in entropy, indicating increased awareness of adversarial uncertainty. Compared to Dropout, SNN produces consistently higher entropy, signaling more cautious predictions. Relative to EDL, SNN is less sensitive. Overall, these results show that SNN provides a balanced tradeoff between robustness and calibrated uncertainty when confronted with adversarial examples.

## 5    CONCLUSION

We introduced *Subjective Neural Networks (SNNs)*, a Bayesian deep learning framework that embeds subjective trust into neural inference. By combining Beta–Bernoulli Dropout with subjective logic, SNNs yield predictions expressed as structured *opinions* that distinguish belief from ignorance. This enables models to communicate not only how confident they are, but also how much uncertainty stems from limited knowledge.

Our experiments on MNIST and CIFAR-5 show that SNNs achieve accuracy comparable to Monte Carlo Dropout and Evidential Deep Learning while providing better-calibrated uncertainty, smoother accuracy–uncertainty tradeoffs, and improved robustness to covariate shift and adversarial perturbations. These results demonstrate that SNNs deliver both strong predictive performance and meaningful uncertainty estimates.

Beyond empirical gains, SNNs highlight a conceptual advance: uncertainty in neural networks should be interpreted, not just quantified. By explicitly encoding subjective trust, they move deep learning closer to systems that can reason about their own reliability, an essential step for trustworthy AI in safety-critical applications.

## REPRODUCIBILITY STATEMENT

We have taken several steps to ensure the reproducibility of our results. All datasets used in our experiments (MNIST, CIFAR-10, and notMNIST) are publicly available, and preprocessing details are provided in Section 4 and the appendix. Architectures, training settings (optimizer, learning rate, batch size, number of epochs), and inference procedures (nested sampling for SNNs, Monte Carlo Dropout passes, FGSM attack settings) are fully specified in Section 3 and Section 4. Additional experimental figures and results are included in the appendix for completeness. An anonymized implementation of our model and training code is provided in the supplementary material to facilitate verification and further experimentation.

## DISCLOSURE OF LLM USAGE

In accordance with the ICLR 2026 policy on Large Language Models, we disclose our use of LLMs in the preparation of this paper. We employed an LLM solely as a general-purpose writing assistant, limited to polishing text for grammar. The model was not used for research ideation, experiment design, analysis, or drafting of technical content. All conceptual contributions, methods, experiments, and results presented in this paper are entirely the authors' own work.

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

# A  ADDITIONAL RESULTS ON CIFAR-5

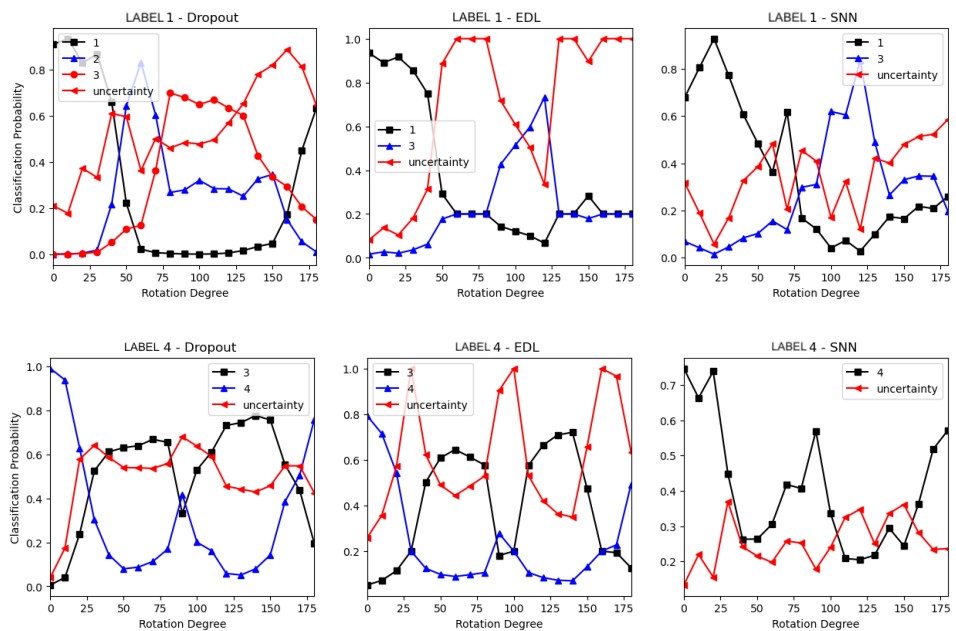

Figure 5: Classification of rotated CIFAR-5 samples across different angles (0°–180°) for Dropout, EDL, and SNN. Rotations cause label switching, and uncertainty varies across methods.

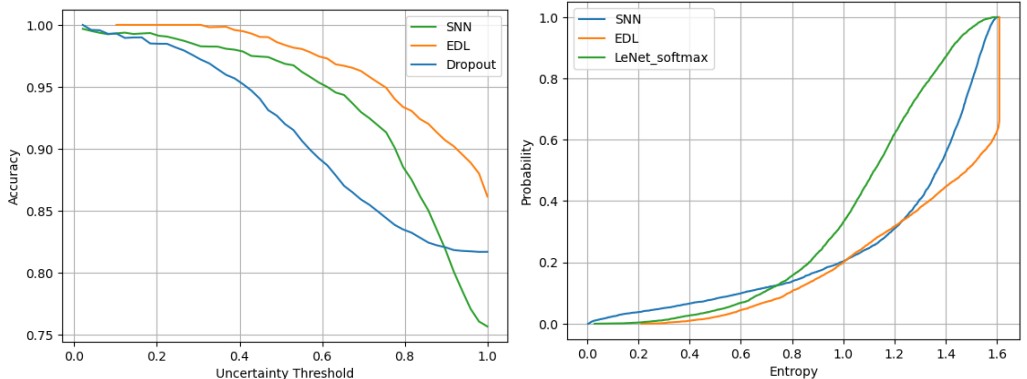

Figure 6: Accuracy vs. uncertainty threshold on CIFAR-5 for Dropout, EDL, and SNN.

Figure 7: Empirical CDF of predictive entropy on OOD data for CIFAR-5.

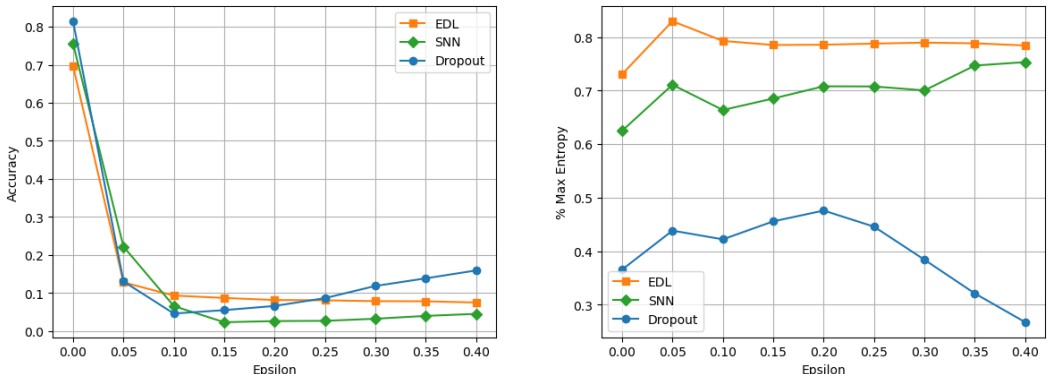

Figure 8: Accuracy and entropy as a function of the adversarial perturbation $\epsilon$ on the CIFAR-5 dataset.

