# OpenReview forum: "Subjective Neural Networks: Bayesian Dropout with Trust-Aware Opinions"
_ICLR.cc/2026/Conference — ICLR 2026 Conference Withdrawn Submission_

### Official Review · Reviewer_273u · 2025-10-31

**Soundness:** 1
**Presentation:** 1
**Contribution:** 1
**Rating:** 2
**Confidence:** 3

**Summary:**

This paper introduce the Subjective Neural Network (SNN), a framework that combines Bayesian variational inference with subjective logic.

**Strengths:**

This paper proposed a novel framework to unify the Bayesian Variational Inference with subjective logic.

**Weaknesses:**

The proposed method is mainly evaluated on small-scale datasets, which makes it hard to convey the effectiveness of the proposed model.

The motivation is not strong, as to why we need to combine the Bayesian Variational Inference with subjective logic.

The experiment result is weak.

**Questions:**

My main concern lies in why you want to combine the Bayesian variational inference with subjective logic.

---

> ### Author Response · Authors · 2025-11-12
> **Motivation for Combining Bayesian Variational Inference and Subjective Logic**
>
> ### **1. Motivation for Combining Bayesian Variational Inference and Subjective Logic**
>
> Subjective Logic allows decomposition of uncertainty into belief and ignorance, which classical Bayesian methods cannot represent explicitly.
>
> Inspired by *DeepTrust*, which somehow assesses how trustworthy a neuron is, we interpret trust in neurons as Beta-distributed latent variables. Bayesian variational inference then provides a principled way to learn these Beta parameters, enabling a nested inference process in which sampled trust probabilities yield Bernoulli masks, and aggregated outputs form Dirichlet opinions.
>
> In essence, the use of Beta distributions enables the model to represent a probability over probabilities, thereby capturing two distinct degrees of uncertainty within a unified formulation.
>
> ---
>
> ### **2. Summary of Planned Improvements**
>
> In the final version, we will:
>
> - Clarify that SNN aims for better and interpretable uncertainty, not higher raw accuracy than MC Dropout.
> - Clarify Figures 1–2 and include Dropout comparisons where relevant.
> - Add computational time analysis and extend experiments to deeper architectures.
> - Add plot of all digits for MNIST and all labels for CIFAR 10
> - Strengthen the motivation with explicit references to trust-aware inference and prior work.
>
> ---
>
> We appreciate the reviewers’ detailed comments, which will help us further improve both the clarity and empirical depth of the paper.
> We believe these clarifications can resolve key misunderstandings and strengthen the paper’s contribution to the field of trust-aware Bayesian deep learning.

---

### Official Review · Reviewer_3Nub · 2025-11-01

**Soundness:** 2
**Presentation:** 2
**Contribution:** 1
**Rating:** 2
**Confidence:** 5

**Summary:**

The authors study uncertainty estimation for image classification. They propose Subjective Neural Networks (SNNs), an approach that combines Beta–Bernoulli Dropout with subjective logic.

They evaluate the approach on MNIST and CIFAR-5 using small CNNs, comparing with MC-dropout and EDL.

**Strengths:**

- The paper is well written in the sense that it contains basically no typos or similar issues.
- The proposed approach is simple and makes some sense overall.

**Weaknesses:**

- The paper is just ~7.5 pages long.
- The experimental evaluation is extremely limited, the proposed approach is just evaluated on MNIST and CIFAR-10 using very small CNNs.
- Even in the very limited evaluation, the proposed method does not seem to perform particularly well, the predictive performance on CIFAR-5 is significantly below MC-dropout in Table 1.

**Questions:**

- Could you please extend the paper to 9 full pages?
- Could you please significantly extend the experimental evaluation with up-to-date datasets and models?

---

> ### Author Response · Authors · 2025-11-12
> **Clarification of Goal and Motivation, Scope of Experiments, and paper length**
>
> We thank all reviewers for their thoughtful and constructive feedback. We address the main concerns below.
>
> ---
>
> ### **1. Clarification of Goal and Motivation**
>
> Our primary goal is *not* to outperform Monte Carlo (MC) Dropout in raw accuracy, but to provide **better and more interpretable uncertainty estimates** while maintaining **higher predictive performance than Evidential Deep Learning (EDL)**.
>
> The proposed Subjective Neural Network (SNN) bridges Bayesian Dropout (which models stochastic uncertainty) and EDL (which models evidence) by explicitly incorporating trust at the neuron level through a Beta distribution.
>
> Compared to EDL, SNN achieves higher accuracy (99.21% vs 98.42 on MNIST and 76.21% vs 71.24%) and smoother uncertainty–accuracy trade-offs (Figures 2–3), while providing uncertainty estimates better than MC Dropout, especially under dataset shift (as illustrated in Figure 1 with rotated images).
>
> We will revise the paper to make this motivation explicit.
>
> ### **2. Scope of Experiments**
>
> We acknowledge that our current experiments use small CNNs on MNIST and CIFAR-10.
> Since SNN introduces a new theoretical formulation (Beta–Bernoulli Dropout + Subjective Logic), we initially validated it on controlled testbeds to isolate and analyze uncertainty behavior.  We also aimed to replicate and do only the evaluation setup used in EDL.
>
> Given the reviewers’ suggestions, we will add evaluations over all digits and CIFAR labels, and extend experiments to deeper architectures (e.g., ResNet) in a revised version.
>
>
> ### **3. Summary of Planned Improvements**
>
> In the final version, we will:
>
> - Clarify that SNN aims for better and interpretable uncertainty, not higher raw accuracy than MC Dropout.
> - Clarify Figures 1–2 and include Dropout comparisons where relevant.
> - Add computational time analysis and extend experiments to deeper architectures.
> - Add plot of all digits for MNIST and all labels for CIFAR 10
> - Strengthen the motivation with explicit references to trust-aware inference and prior work.
>
> ---
>
> We appreciate the reviewers’ detailed comments, which will help us further improve both the clarity and empirical depth of the paper.
> We believe these clarifications can resolve key misunderstandings and strengthen the paper’s contribution to the field of trust-aware Bayesian deep learning.

---

### Official Review · Reviewer_yjur · 2025-11-04

**Soundness:** 2
**Presentation:** 2
**Contribution:** 2
**Rating:** 2
**Confidence:** 4

**Summary:**

The authors of the paper introduce Subjective Neural Network (SNN), a framework that combines Bayesian variational inference with the domain of subjective logic, in order to tackle the problem of uncertainty quantification within the context of deep neural networks (DNNs). Their method adopts principles from Subject logic, in order to make more accurate predictions, namely the representation of beliefs opinions (instead of fixed probabilities), the epistemic uncertainty and prior information.
In order to do so they introduce a framework at which (a) neuron activations are controlled by Beta–Bernoulli Dropout, an extension of classical dropout where, instead of a fixed rate across all neurons, each neuron has a distribution over dropout probabilities (b) the representation of opinion maps to a Dirichlet distribution. Given that direct inference of the posterior over latent trust probabilities is intractable, they adopt a variational approach and also employ the Kumaraswamy distribution, in order to enable gradient-based learning.
At inference time, they approximate the posterior predictive distribution through nested sampling.
They conclude their work with an experiments section on CIFAR-10 and MNIST data where they test the performance of their method against Dropout and Evidential Deep Learning (EDL).

**Strengths:**

The authors introduce the idea of combining two frameworks, namely Bayesian variational inference with subjective logic.
They develop at which dropout probabilities for each neuron are not fixed, but rather stem from a hierarchical distribution ( $z_j \sim Beta(a_j, b_j),   \ p_j \sim Bernoulli(z_j), \foreach j $)
In order to avoid the problem of intractable posterior they use they approximate each $p_j$ with a variational Beta distribution and enable gradient learning through Kumaraswamy distribution which approximates the Beta.
Finally, they use nested sampling for inference.

**Weaknesses:**

-- The introduction of the paper has no references;
-- Although the idea is interesting, the method does not seem to outperform existing ones at the experiments section (eg. Table 1);
-- Figure 1 is not very clear; moreover dropout is tested against different sets of digits at the digit 9 case;
-- Figure 2 does not include Dropout;
-- In CIFAR case (which is presented at the Appendix) SNN is outperformed across all tests (also same problem is observed as at Figure 5, methods are not tested against the same set of labels)

**Questions:**

Authors would be kindly asked to

1) please provide computational times of Dropout, EDL and SNN;
2) please include more results with respect to MNIST and CIFAR (more digits and labels at the classification of rotated digits and samples, respectively;
3) please regenerate Figure 1, bottom row so that (a) all methods are tested against the same set of digits; (b) symbols are consistent to digits across graphs;
4) please regenerate Figure 2, including Dropout curve

---

> ### Author Response · Authors · 2025-11-12
> **Clarification of Goal and Motivation, Figures 1 and 2, and Computational Complexity**
>
> We thank all reviewers for their thoughtful and constructive feedback. We are encouraged that the idea of combining Bayesian Variational Inference with Subjective Logic (SL) was found interesting, and we address the main concerns below.
>
> ---
>
> ### **1. Clarification of Goal and Motivation**
>
> Our primary goal is *not* to outperform Monte Carlo (MC) Dropout in raw accuracy, but to provide **better and more interpretable uncertainty estimates** while maintaining **higher predictive performance than Evidential Deep Learning (EDL)**.
>
> The proposed Subjective Neural Network (SNN) bridges Bayesian Dropout (which models stochastic uncertainty) and EDL (which models evidence) by explicitly incorporating trust at the neuron level through a Beta distribution.
>
> Compared to EDL, SNN achieves higher accuracy (99.21% vs 98.42 on MNIST and 76.21% vs 71.24%) and smoother uncertainty–accuracy trade-offs (Figures 2–3), while providing uncertainty estimates better than MC Dropout, especially under dataset shift (as illustrated in Figure 1 with rotated images).
>
> We will revise the paper to make this motivation explicit.
>
> ### **2. Clarification of Figures 1 and 2**
>
> - **Figure 1 (rotated digits):**
>   This figure visualizes the predicted probabilities for each model across rotation angles of digit 9 (and 1). The digits shown below the plots correspond to predicted labels, not ground truth. We will clarify this in the caption.
>
> - **Figure 2:**
>   This figure omits MC Dropout because it follows the evaluation protocol of EDL (Sensoy et al., 2018), where only EDL is evaluated.
>   In the revised version, we will include the MC Dropout curve for completeness.
>
> ---
>
> ### **3. Computational Complexity**
>
> The inference computational complexity of SNN arises from the nested inference procedure with \\( N_p \\) Beta samples and \\( N_m \\) Bernoulli masks.
>
> For the evaluation, we set:
> \\[
> N_{\text{MC}} = 100 \quad \text{for MC Dropout, and} \quad N_p \times N_m = 10 \times 10 = 100 \quad \text{for SNN,}
> \\]
> ensuring identical numbers of forward passes and thus comparable computational cost.
> We will make the computational complexity clear in Section 4.
>
> For completeness, note that EDL requires only a single forward inference pass.
>
>
> ### **4. Summary of Planned Improvements**
>
> In the final version, we will:
>
> - Clarify that SNN aims for better and interpretable uncertainty, not higher raw accuracy than MC Dropout.
> - Clarify Figures 1–2 and include Dropout comparisons where relevant.
> - Add plot for all digits for MNIST and all labels for CIFAR 10
> - Add computational time analysis and extend experiments to deeper architectures.
> - Strengthen the motivation with explicit references to trust-aware inference and prior work.
>
> ---
>
> We appreciate the reviewers’ detailed comments, which will help us further improve both the clarity and empirical depth of the paper.
> We believe these clarifications can resolve key misunderstandings and strengthen the paper’s contribution to the field of trust-aware Bayesian deep learning.

---

### Note · Authors · 2026-01-26

**Comment:**

Dear, thank you for the review. I improved it and need to submit it to another conference.

**Withdrawal Confirmation:**

I have read and agree with the venue's withdrawal policy on behalf of myself and my co-authors.

---

### Meta-Review · Area_Chair_KjmF · 2026-01-03

**Summary:**

The paper seems incomplete with only 7.5 pages while leaving a significant room for improvement. All reviewers agree that the work is not motivated very well, the experiments are limited, and the presentation requires improvements. The author rebuttal does not fully resolve any of these concerns. Making a lot of promises for post-rebuttal updates does not guarrantee resolution of any of these concerns and therefore, the AC recommends rejection of the paper.

The manuscript appears premature for publication, spanning only 7.5 pages and leaving substantial room for improvement. The reviewers reached a consensus that the work lacks strong motivation, the experimental evaluation is too narrow, and the overall presentation is insufficient. While the authors provided a rebuttal, it failed to resolve these core issues; furthermore, the AC notes that promises of extensive post-rebuttal updates cannot guarantee a high-quality final version. Consequently, the AC recommends rejection.

**Reviewer Concerns:**

Most concerns remain unresolved. The authors promises many updates but even with these additional contents, it is unclear if they would address the concerns raised.

**Reviewer Scores:**

The current three 'Reject' scores are unlikely to be upgraded, as most concerns remain unresolved.

---

### Decision · Program_Chairs · 2026-01-26

Reject